# Heterogeneity between Core Needle Biopsy and Synchronous Axillary Lymph Node Metastases in Early Breast Cancer Patients—A Comparison of HER2, Estrogen and Progesterone Receptor Expression Profiles during Primary Treatment Regime

**DOI:** 10.3390/cancers14081863

**Published:** 2022-04-07

**Authors:** Laura Weydandt, Ivonne Nel, Anne Kreklau, Lars-Christian Horn, Bahriye Aktas

**Affiliations:** 1Department of Gynecology, Medical Center, University of Leipzig, 04103 Leipzig, Germany; ivonne.nel@medizin.uni-leipzig.de (I.N.); anne.kreklau@medizin.uni-leipzig.de (A.K.); bahriye.aktas@medizin.uni-leipzig.de (B.A.); 2Department of Pathology, Medical Center, University of Leipzig, 04103 Leipzig, Germany; lars-christian.horn@medizin.uni-leipzig.de

**Keywords:** tumor heterogeneity, receptor change, breast cancer, core needle biopsy, axillary lymph node metastases, human epidermal growth factor receptor-2, estrogen, progesterone, Ki67, receptor expression profiles

## Abstract

**Simple Summary:**

All initial therapeutic decisions in early breast cancer are commonly based on the intrinsic subtype consisting of estrogen (ER), progesterone (PR), the human epidermal growth factor 2 (HER2) receptors and the proliferation marker Ki67. However, breast cancer is a very heterogeneous disease, and receptor expression is reported to change during progression. Little is known about receptor changes at the primary site. In a German single center study, we retrospectively analyzed a mostly therapy naive cohort of 215 primary breast cancer patients with axillary synchronous lymph node metastases (LNM). We compared core needle biopsy tissue of the primary tumor (t-CNB) to axillary LNM and detected receptor discordance for all three receptors at the primary site.

**Abstract:**

In breast cancer therapeutic decisions are based on the expression of estrogen (ER), progesterone (PR), the human epidermal growth factor 2 (HER2) receptors and the proliferation marker Ki67. However, only little is known concerning heterogeneity between the primary tumor and axillary lymph node metastases (LNM) in the primary site. We retrospectively analyzed receptor profiles of 215 early breast cancer patients with axillary synchronous LNM. Of our cohort, 69% were therapy naive and did not receive neoadjuvant treatment. Using immunohistochemistry, receptor status and Ki67 were compared between core needle biopsy of the tumor (t-CNB) and axillary LNM obtained during surgery. The discordance rates between t-CNB and axillary LNM were 12% for HER2, 6% for ER and 20% for PR. Receptor discordance appears to already occur at the primary site. Receptor losses might play a role concerning overtreatment concomitant with adverse drug effects, while receptor gains might be an option for additional targeted or endocrine therapy. Hence, not only receptor profiles of the tumor tissue but also of the synchronous axillary LNM should be considered in the choice of treatment.

## 1. Introduction

Breast cancer is a very heterogeneous disease and can be classified into distinct intrinsic subtypes based on the expression profiles of the estrogen receptor (ER), progesterone receptor (PR), the extent of human epidermal growth factor receptor 2 (HER2) and Ki67 index. Patients diagnosed with ER and/or PR positive tumors often receive endocrine therapy. Patients suffering from HER2 amplified tumors are prone to treatment with targeted therapy such as trastuzumab and pertuzumab. Triple negative breast cancer (TNBC), is hormone-receptor negative (neither ER nor PR) and HER2 negative with poor prognosis. The majority of TNBCs show an aggressive phenotype. Patients usually receive chemotherapy but have an increased risk of recurrence. Immunohistochemical staining of ER, PR and HER2 in addition to the proliferation marker Ki67 are the basis for classifying breast cancer tumors according to the St. Gallen subtype classification into five different intrinsic subtypes: luminal A, luminal B/HER2 positive, luminal B/HER2 negative, HER2 enriched and triple negative. Receptor status profiling upfront surgical treatment is decisive when primary systemic treatment such as neoadjuvant chemotherapy (NACT) is required. Here, pre-surgical core needle biopsy tissue of the primary tumor (t-CNB) is a valuable diagnostic tool to determine ER, PR and HER2 expression. Concordance of ER, PR and HER2 profiles between t-CNB and surgical tumor specimens were described to be adequate with rates of 93.6%, 85.9%, and 96.3%, respectively [1,2].

Commonly, a tumor is considered hormone receptor (HR) positive if at least 1% of the tumor cells appear visible using immunohistochemical staining against ER and/or PR [3]. Recently, the updated guidelines for treatment with endocrine therapy revealed different cut-off values varying between ≥1% and ≥10% positive cells. Retrospective analysis has found that patients with 1–9% ER/PR positive tumor cells, so called low-positive HR patients, had worse survival rates compared to patients with ER/PR ≥ 10% [4,5]. Interestingly, it was reported that ER low-positive tumors seemed to behave similarly to the aggressive triple negative tumor phenotype in terms of chemotherapy response rates and treatment resistance [6,7,8]. Yet, it is unclear whether this also applies to PR low tumors. Profiles of ER, PR and HER2 expression appeared to be inconsistent between the primary tumor and the axillary lymph nodes or metastatic spreads [9]. This receptor discordance has gained increased attention particularly regarding the treatment of patients with metastatic breast cancer. Numerous studies have indicated a major role during tumor progression and hence in patients’ prognosis [10]. Particularly, discordance between the primary and nodal disease was reported to be correlated with a poor prognosis [11]. In women with luminal A breast cancer, a switch to non-luminal A lymph node metastasis was described in 38% of the cases in a retrospective study with 146 patients [12]. The inconsistent molecular subtype was correlated with aggressiveness and poor prognosis, thus suggesting systemic treatment with chemotherapy. Furthermore, altered receptor expressions between the primary tumor and lymph node metastases might play a role in acquired resistance during adjuvant targeted or endocrine therapy.

Axillary surgery has long been an established part of the management of primary breast cancer. While there have been many discussions about the extent of axillary surgery and classifications of axillary lymph nodes [13], either sentinel lymph node biopsy (SLNB) for clinically and radiologically uninvolved axilla, or complete axillary dissection (ALND) in cases of axillary lymph node metastases (LNM) is common practice [14].

However, only little research data has been published concerning heterogeneity between primary tumor and synchronous axillary lymph node metastases (LNM) during primary treatment [15,16,17,18]. It remains unclear whether receptor changes preferentially occur during disease recurrence and metastatic progression or whether they already appear at the beginning of the disease, before any systemic treatment has been applied. Here we present a comprehensive comparison of receptor expression profiles between pre-surgical t-CNB and synchronous axillary lymph node metastases at primary diagnosis, revealing therapeutically relevant aspects of tumor heterogeneity in breast cancer patients from a German single-center study.

## 2. Materials and Methods

This retrospective study was approved by the ethics committee of the medical faculty of the University of Leipzig (internal reference no. 181/20-ek). We included 215 patients, who were diagnosed with primary breast cancer and synchronous axillary LNM. All patients received treatment between 2008 and 2020 at the Department of Gynecology, Medical Center, University of Leipzig, Germany (Table 1). The median age was 61 years, ranging from 31 to 85 years. The majority of the patients (74%) was diagnosed with an invasive carcinoma of no special type, also termed as ductal invasive carcinoma, 18 (8.4%) patients had an invasive lobular carcinoma, 33 (15.3%) patients suffered from a mixed invasive ductal-lobular carcinoma and 15 (7%) patients had a pre-surgical carcinoma and a post-surgical pTis tumor stage. Most patients had a primary tumor stage of pT1 (43.7%) or pT2 (34.9%), whereas an advanced local disease with pT3 and pT4 was diagnosed in only 23 (10.7%) and 8 (3.7%) patients, respectively. The final nodal status was pN0 after neoadjuvant chemotherapy in 19 (8.8%) patients, pN1 in 129 (60%) patients, pN2 in 50 (23.2%) and pN3 in 14 (8%) of the cases. Primary surgery (either sentinel lymph node biopsy (SLNB): 54.9% or axillary lymph node dissection (ALND): 45.1%) was performed in 149 (69.3%) cases without any neoadjuvant therapy; therefore, the majority of our patient cohort was therapy naive. Neoadjuvant chemotherapy was applied in 46 (21.4%) cases and neoadjuvant endocrine therapy in 20 (9.3%) cases. Concerning the grading, 57 (26.5%) cases showed G1, 115 (53.5%) G2 and 43 (20%) G3 in the t-CNB. For disease management and patient prognosis, the clinicopathological characteristics and the receptor expression profiles of t-CNB specimens were transmitted into molecular intrinsic subtypes according to the St. Gallen classification [19]. This stratification revealed 41.4% luminal A tumors, 11.6% luminal B/HER2 positive tumors, 26.5% luminal B/HER2 negative tumors, 6.5% HER2 enriched tumors and 7.9% triple negative tumors.

The basic patient characteristics and clinicopathological data, including age, tumor stage, grading, nodal status, histological subtype, receptor status (ER, PR, HER2) and Ki67 index of the t-CNB were received during clinical routines and documented in the medical reports. For our study, we performed a re-analysis of the Ki67 index and expression profiling of ER, PR, HER2, using formalin-fixed, paraffin-embedded (FFPE) specimens from synchronous axillary LNM. The material was obtained by core needle biopsy at primary diagnosis (t-CNB) of the breast and during primary oncological surgery of the axilla (LNM), respectively. Sentinel lymph nodes were identified with preoperative injection of Tc99. In cases of multiple positive lymph nodes occurring during surgery, we chose one specimen for immunohistochemical staining and evaluation.

Immunohistochemical examination was performed according to the ASCO/CAP guidelines using the Ventana platform [3,20,21,22]. Due to recurring discussions in the literature concerning different cut-off values for ER and PR to initiate endocrine therapy [4], stained specimens were divided into two groups with cut-off levels at either ≥1% or ≥10%. A tumor was defined as ER/PR positive if expression was observed in at least 1% and 10% of the tumor cells, respectively, and ER/PR negative when <1% and <10% of the tumor cells, respectively, were positive. The immunohistochemical evaluation of HER2 was performed by the local pathology department according to the most recent ASCO-guidelines [22]. All the cases were evaluated by the same experienced pathologist who examined all the specimens for this study.

Membranous HER2-staining was scored as follows: Negative = 0 (no staining) and 1+ (weak incomplete staining in <10% tumor cells); Equivocal = 2+ (weak/moderate complete staining in at least 10% of the tumor cells); Positive = 3+ (strong complete membrane staining in at least 30% of the tumor cells). In the case of equivocal (2+) HER2-immunostaining, additional chromogenic in situ hybridization (CISH) analysis of the primary breast tumor was performed [22]. Patients were considered HER2 negative when immunostaining was equivocal (2+) and CISH was negative. In four cases the immunostaining against HER2 was equivocal (2+) in the axillary LNM. Due to limited tissue no additional CISH was performed, and therefore, we interpreted the receptor as stable compared to the t-CNB.

After final evaluation of ER, PR, HER2 and Ki67 immunohistochemical staining results, LNM specimens were grouped into intrinsic subtypes (Luminal A, Luminal/B HER negative, Luminal B/HER2 positive, HER2 enriched and triple negative) according to St Gallen classification [19] and compared with intrinsic subtypes of the t-CNB.

Statistical data analysis was performed using SPSS version 21 (IBM Corp, Armonk NY, USA). Normality of continuous data was tested by means of the Kolmogorov–Smirnov test and the homogeneity of variances was analyzed using Levene’s test. Descriptive statistics were used to compare the baseline characteristics between patients according to the presence of receptor heterogeneity and tumor stage, tumor grading and neoadjuvant therapy regime, while median and interquartile range (IQR) were used for continuous variables. A *p*-value of < 0.05 was used as the threshold of statistical significance for all analyses. Inclusion criteria for the analysis was a complete data set for each patient. Patients with missing data were treated as missing completely at random and excluded from specific analysis.

## 3. Results

### 3.1. Comparison of ER, PR and HER2 Status in t-CNB and LNM Using ≥1% vs. ≥10% Positivity Cut-Off Levels

We analyzed receptor expression profiles as described above in 215 t-CNB and 211 matching axillary LNM using two different cut-off values. Overall, four LNM were not available for analysis due to a lack of spare tumor tissue.

When comparing ER positivity rates between both cut-off groups, the shift from ≥10% to ≥1% positive cells appeared to have no discernable impact concerning the receptor status of the t-CNB and LNM. In the t-CNB, ER receptor status was very similar with positivity rates as high as 83.7% and 83.3% in the ≥1% and ≥10% groups, respectively. When looking at the receptor profiles of the axillary LNM, ER positivity was again very similar with rates of 78.1% and 76.7% in the ≥1% and. ≥10% groups, respectively.

However, considering the PR profile, decreasing the cut-off level from ≥10% to ≥1% positive cells revealed a statistically significant increase of the PR positivity rate among t-CNB specimens and particularly among LNM samples. In t-CNB samples, 153 and 166 tumors were PR positive using ≥10% and ≥1% cut-off values, respectively, indicating a 6% increase. The number of positive cases increased by 13 when using the ≥1% cut-off compared to ≥10%. The rate of PR positive tumors increased by 12.6% (105 vs. 132) in LNM specimens when comparing ≥10% vs. ≥1% cut-off values (*p* < 0.001).

HER2 was positive in 19.1% of the t-CNB cases and in 10.7% of the LNM specimens. Hence, the HER2 positivity appeared to drop in nearly half of the cases when comparing t-CNB to LNM (*p* < 0.01).

The expression profiles are summarized in Table 2 and representative images of immunohistochemical evaluation are shown in Figure 1.

### 3.2. Analysis of ER, PR and HER2 Discordance between t-CNB and LNM

We compared receptor expression profiles of ER, PR and HER2 between t-CNB and 211 notable matching axillary LNM and analyzed receptor changes. A receptor change occurred either as a receptor loss (t-CNB receptor positive and matching LNM receptor negative) or as a receptor gain (t-CNB receptor negative and matching LNM receptor positive). In total, 13 cases showed a receptor gain in LNM compared to t-CNB and a receptor loss was found in 69 cases when using the ≥1% cut-off (Table 3). When the ≥10% cut-off was applied, the number of specimens with receptor gain was 13, similar to with the decreased cut-off value. However, the number of receptor losses was increased. Hence, when applying the ≥10% cut-off, the total number of detected receptor losses increased to 85.

As shown in Table 3, the discordance rates between axillary LNM and t-CNB were 6% and 6.9%, respectively, for ER, 20% and 26.5%, respectively, for PR in the ≥1% and ≥10% groups, and 12.1% for HER2 (Figure 2). We observed an increased occurrence of receptor changes for both ER and PR when applying the ≥10% cut-off. On the other hand, discordance rates were decreased when applying the ≥1% cut-off. In our analysis, PR was the receptor which changed most frequently. The overall discordance rates are shown in Figure 2.

### 3.3. Analysis of Intrinsic Subtype Discordance between LNM and t-CNB (Using ≥1% Cut-Off Value)

Additionally, we compared intrinsic subtypes between t-CNB and LNM. Descriptive analysis of the intrinsic subtypes in the LNM specimens showed an increase of luminal A tumors up to 50.2%, a decrease in luminal B/HER2 positive tumors to 7%, a decrease in luminal B/HER2 negative tumors to 19.5%, a drop of HER2 enriched tumors down to 3.7% and an increased rate of triple negative tumors as high as 13.5% (Figure 3). It is noteworthy that receptor losses were associated with a decreased proliferation index in LNM, possibly causing an increase of luminal A specimens (Table 4).

### 3.4. Correlation between Clinicopathological Parameters and Receptor Heterogeneity

Competitive analysis of tumor associated variables and receptor heterogeneity using the ≥1% receptor positivity cut-off in LNM compared to t-CNB revealed that a change (loss or gain) of ER was by trend associated with an increased Ki67 index (40% vs. 10% in the no change group; *p* = 0.11) and occurred significantly more often in younger patients (median 53 vs. 62 years in the no change group; *p* < 0.01). When considering the PR change using the ≥1% positivity specimens of LNM and t-CNB, although not significantly, a change appeared to be associated with patients younger than 63 years. It is worth noting that a HER2 change, which was mostly a loss of staining, was significantly correlated with a decreased Ki67 index (15 vs. 3 in the no change/change group; *p* = 0.01) in t-CNB (Table 4), indicating a possible switch towards luminal A tumor characteristics and presenting a contrast to the increase of more aggressive triple negative tumors.

We analyzed different tumor stages between groups with vs. without receptor changes. As for PR, we noticed a significant correlation between tumor stage and receptor heterogeneity, although with contradictory figures within the subgroups. PR changes were decreased by 7% among patients with T1, but increased by 6% among T2 tumors. In patients with T3 tumors we did not observe PR changes. Among patients with T4 and Tis, PR heterogeneity was increased by a factor of four and two, respectively. For ER and HER2, heterogeneity was not significantly associated with tumor stage. Hence, there was no definite correlation between receptor heterogeneity and tumor stage.

Further, we did not find any significant associations between a receptor change (ER; PR; HER2) and therapy-related parameters.

The majority of our cohort had not received any upfront therapy. However, a small sub-cohort of 46 patients received NACT and 20 patients were treated with endocrine therapy. When examining ER, we found an increased rate of changes in the NACT-group compared with patients without therapy or with endocrine treatment (8.9% vs. 5.5% vs. 5.0%). Further, we found an increased number of HER2 changes when comparing no therapy and NACT (10.7% vs. 22.7%, no statistical significance). The number of PR changes increased in patients treated with endocrine therapy compared to the no therapy and NACT groups (35% vs. 18.6% vs. 20%, no level of statistical significance, Table 4).

## 4. Discussion

In this study we made extensive comparisons between receptor expression profiles of pre-surgical t-CNB of the primary tumor, and synchronous axillary lymph node metastases (LNM) obtained during surgery of early breast cancer patients within primary treatment regimes. ER, PR and HER2 status was immunohistochemically analyzed using the common cut-off at ≥10% positive tumor cells vs. the much debated low-positive cut-off at ≥1%. Overall, 82 of 645 receptors had at least one change using the ≥1% cut-off value. This finding is consistent with a study by Aitken et al. showing a significant number of patients with discordant quantitative expression of molecular markers between primary and nodal disease. They reported that 46.9% of the analyzed cases had disparate breast/node receptor status of at least one receptor [23]. Additionally, other research groups also reported high discordance rates between synchronous axillary LNM and the primary tumor, although with lower numbers of matched pairs [24]. When we split the positive ER/PR groups into low and high using cut-off levels ≥1% vs. ≥10% positive tumor cells, no major differences were observed in the ER status. The rate of PR positive tumors, however, increased by 12.6% in the LNM when the cut-off was decreased to ≥1%, indicating that these patients might have the additional option to receive endocrine therapy in case they were not already ER positive. We focused on receptor changes in LNM specimens obtained during surgery compared with pre-surgical t-CNB at primary diagnosis. Discordance rates were 6% for ER, 20% for PR and 12.1% for HER2 between LNM and t-CNB (using a cut-off of ≥1%). The discordance rates appear to match analyses of distant metastases and primary breast tumor tissue which were reported to vary between 7 and 50% for ER, 10 and 50% for PR and 3 and 30% for HER2 [11,25,26,27,28,29]. It is widely accepted that receptor expression might change during tumor progression [10]. Hence, at recurrence, tissue biopsy and reassessment of biological as well as immunohistochemical features has become a standard procedure in the treatment of breast cancer [21].

Our data, however, suggest that at primary diagnosis, a receptor profile assessment not solely based on the t-CNB, but additionally on the synchronous axillary LNM, might draw a clearer picture of the tumor before treatment decisions are made.

In our cohort, we observed a higher number of patients with receptor losses than with receptor gains. In total we found 69 losses vs. 13 gains using a cut-off of ≥1%. Hence, our data strengthen the assumption that receptor loss seems to be a more frequent event than receptor gain [30]. Furthermore, the discordance rates of receptor profiles in LNM compared with t-CNB indicated that PR appeared to be the most unstable. This confirms reported data from de Duenas et al. [31]. Various hypotheses were postulated to explain receptor discordance. One explanation for possible receptor changes between synchronous LNM and primary tumor might be the clonal selection hypothesis, which suggests that the primary tumor consists of multiple clonal subpopulations that are capable of forming metastases and result in different tumor cell phenotypes [32,33,34]. Concerning the hormonal receptors ER and PR, we observed that changes occurred more frequently in younger patients, underlining the evidence that younger patients tend to suffer from more aggressive tumors and thus have a poorer prognosis [35]. Our results suggest that metastatic cells seek to become triple negative and hence eliminate their targetable receptors in order to obtain a more aggressive phenotype. This is further supported by the increased proportion of the triple-negative intrinsic subtype and decreased proportions of HER2-enriched and luminal B tumors in LNM compared with t-CNB specimens. On the other hand, it may be possible that triple negative cells are the more motile cancer cells and thus metastasizing to the lymph nodes. Surprisingly, HER2 changes, which were predominantly receptor losses, were associated with decreased cell proliferation in the t-CNB specimens, indicating that luminal B/HER2 tumors might eliminate HER2 and hence gain a luminal A phenotype, which is consistent with decreased Ki67. Accordingly, we noticed an increase in luminal A tumors (41.4% in t-CNB vs. 50.2% in LNM) that are associated with a good prognosis [36] and require no further chemo- but endocrine therapy. On the one hand, based on the current guidelines, these patients might receive overtreatment with the risk of adverse drug effects. On the other hand, patients with tumors that present a triple-negative phenotype in the nodal sample might not receive appropriate treatment with chemotherapy. Considering that we observed both an increase in luminal A tumors with good prognosis and less therapeutic necessity as well as an increase in aggressive triple negative tumor subtypes that require additional therapy, it is crucial to detect alterations of the receptor expression profiles in the LNM compared with t-CNB.

While other research groups suggested the selective pressure of therapy might lead to a receptor change [37], we could not entirely confirm this, as 69.3% of our cohort had no upfront therapy. It is important to emphasize that the majority of our patients had not received any therapy, yet our data revealed discordance between postsurgical LNM tumor specimens and receptor profiles obtained from pre-surgical t-CNB. Hence, possible theories concerning the influence of therapy seem not very likely. Here, the theory of intra-tumoral heterogeneity might be applicable. This theory suggests that a single biopsy is unlikely to represent the genomic landscape of a patient’s cancer accurately [38,39,40]. The variable expression of biomarkers within a heterogeneous tumor such as breast cancer might cause interpretation problems and lead to discordant results in small biopsies such as t-CNB [41].

Hence, a receptor gain might occur in cases when a sub-clonal tumor cell population was missed during small biopsies of t-CNB but positively analyzed in the surgical tumor sample of LNM, which usually delivers sufficient tissue amounts.

Other research groups have shown that a discordance in the receptor profile may have an impact on the patient’s outcome [11]. Patients who had PR negative primary tumors and PR positive paired LNM had higher survival rates when treated with endocrine therapy compared with those not receiving endocrine therapy. Additionally, patients with a detected HER2 heterogeneity had a decreased disease-free survival [42]. At this time, we cannot yet answer outcome related questions as we lack survival data for our patients.

Although our study included a high number of patients without any upfront therapy, there was also a small sub-cohort of patients receiving neoadjuvant therapy. Among this sub-cohort, we observed an increased rate of receptor changes. In the sub-cohort treated with endocrine therapy, we discovered an elevated rate of PR changes. Despite the small number of patients in the sub-groups, these findings might add strength to the theory that therapy causes selective pressure, which could lead to a receptor change, in most cases a receptor loss. In order to investigate receptor changes during primary treatment, studies with larger patient cohorts are required.

Since our study was a retrospective analysis, no change of treatment was performed. Especially receptor gains might offer a new possible target for further therapies. Hence, guidelines recommend to consider a targeted therapy when receptors are positive at least in one biopsy, regardless of the location [21]. Further research needs to be conducted to investigate whether an individual treatment decision based on the LNM and t-CNB receptor profile at primary diagnosis might improve clinical outcome of breast cancer patients.

## 5. Conclusions

Receptor discordance between t-CNB and synchronous axillary LNM appears to already exist at the primary site and might reflect intra-tumor heterogeneity. Hence, receptor profiles of the tumor tissue and the synchronous axillary LNM should be considered in treatment decisions, not only at recurrence but particularly also during the primary treatment regime.

## Figures and Tables

**Figure 1 cancers-14-01863-f001:**
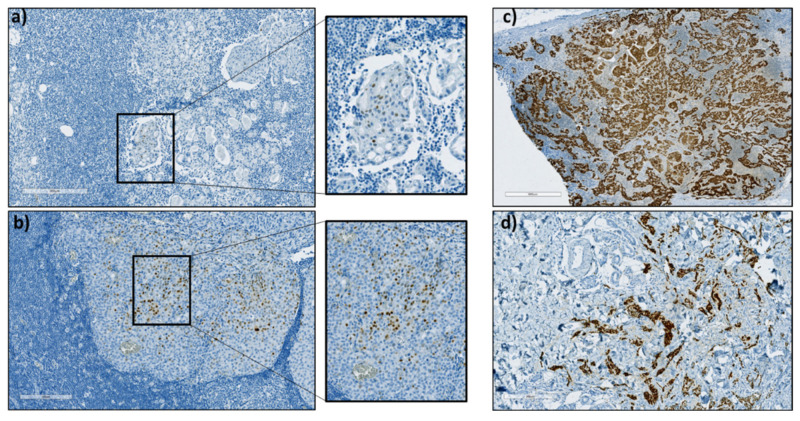
Immunohistochemical staining against PR in LNM specimens showing (**a**) ≥1% PR positive tumor cells, (**b**) ≥10% PR positive tumor cells, (**c**) 100% PR positive tumor cells and (**d**) a t-CNB derived from a PR positive mamma carcinoma which was used as a positive control.

**Figure 2 cancers-14-01863-f002:**
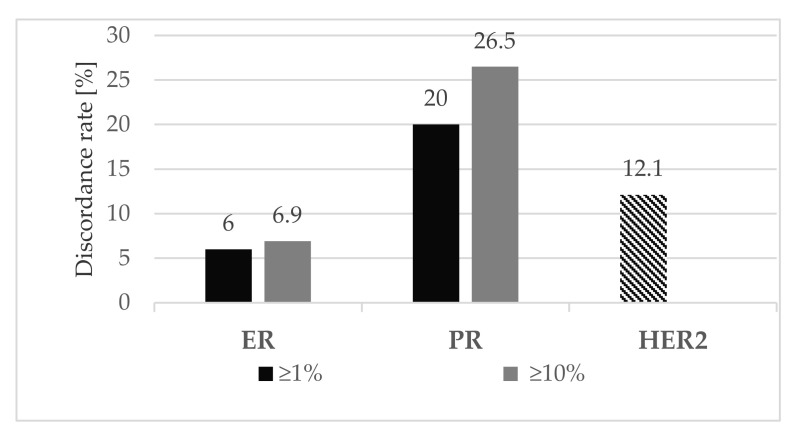
Discordance rates of ER, PR (using cut-offs ≥1% and ≥10%) and HER2 in LNM compared with t-CNB. The discordance rates between axillary LNM and t-CNB were 6% and 6.9%, respectively for ER, 20% and 26.5%, respectively for PR in the ≥1% and ≥10% groups, and 12.1% for HER2.

**Figure 3 cancers-14-01863-f003:**
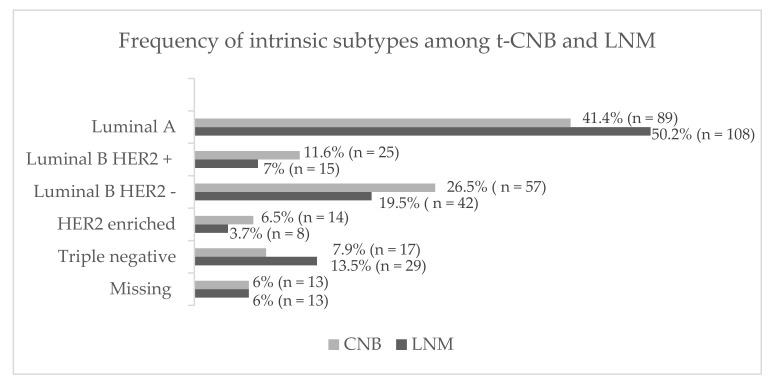
Changes of intrinsic subtypes in t-CNB compared to LNM (*n* = 202). Both, luminal A tumors and triple negative tumors, increased by 8.8% and 5.9%, respectively. The rate of luminal B and HER2 enriched tumors decreased between t-CNB and LNM specimens.

**Table 1 cancers-14-01863-t001:** Baseline characteristics.

Parameters	No. of Patients	
	*n* = 215	%
Median age [years] (IQR)	61 (50–73)	
Premenopausal	55 (25.6)	
Perimenopausal	5 (2.3)	
Postmenopausal	155 (72.1)	
Surgery		
Sentinel node biopsy	118	54.9
Axillary node dissection	97	45.1
Pathological tumor stage		
pTis	15	7
pT1	94	43.7
pT2	75	34.9
pT3	23	10.7
pT4	8	3.7
Pathological nodal status *		
pN0	19	8.8
pN1	129	60
pN2	50	23.2
pN3	17	8
Upfront therapy		
No therapy	149	69.3
Neoadjuvant chemotherapy	46	21.4
Endocrine therapy	20	9.3
t-CNB analyzed	215	100
t-CNB histology		
Invasive carcinoma of no special type	159	74
Invasive lobular	18	8.4
Mixed invasive ductal and lobular	33	15.3
Other histology (e.g., metaplastic, mucinous)	5	2.3
t-CNB grading		
G1	57	26.5
G2	115	53.5
G3	43	20
t-CNB intrinsic subtype		
Luminal A	89	41.4
Luminal B/HER2+	25	11.6
Luminal B/HER2−	57	26.5
HER2 enriched	14	6.5
Triple negative	17	7.9
Not available **	13	6
LNM analyzed	211	98.1
LNM missing ***	4	1.9
Median time [days] between t-CNB and surgery		
Overall	65.6	
No neoadjuvant therapy	23.8	
Neoadjuvant chemotherapy	176.5	
Neoadjuvant endocrine therapy	118.3	

* after final breast surgery ** due to missing Ki67 *** due to no spare tumor tissue.

**Table 2 cancers-14-01863-t002:** Receptor analysis of ER, PR and HER2.

Parameters	t-CNB	LNM	
	*n* (%)	*n* (%)	*p*-Value
ER analyzed	215 (100)	211 * (98.1)	
positive (≥1%)	180 (83.7)	168 (78.1)	0.02
positive (≥10%)	179 (83.3)	165 (76.7)	<0.01
*p*-value	0.25	1	
PR analyzed	215 (100)	211 * (98.1)	
positive (≥1%)	166 (77.2)	132 (61.4)	<0.001
positive (≥10%)	153 (71.2)	105 (48.8)	<0.001
*p*-value	<0.001	<0.001	
HER2 analyzed	215 (100)	205 * (95.3)	
positive	41 (19.1)	23 (10.7)	<0.01

Receptor analysis in t-CNB and LNM specimens using different cut-off values for ER and PR, showing a significant increase of PR positive cases when applying the ≥1% cut-off compared to the ≥10% cut-off. Further, receptor positivity was significantly decreased in the LNM vs. t-CNB specimens. * Some LNM were not available for further receptor analysis due to a lack of spare tumor tissue.

**Table 3 cancers-14-01863-t003:** Change of ER, PR and HER receptor status in t-CNB vs. LNM.

Receptor	Change	Cut-Off ≥ 1%	Cut-Off ≥ 10%
		*n* (%)	*n* (%)
ER	Overall	13(6)	15 (6.9)
	Receptor loss	11 (5.1)	13 (6)
	Receptor gain	2 (0.9)	2 (0.9)
PR	Overall	43 (20)	57 (26.5)
	Receptor loss	37 (17.2)	51 (23.7)
	Receptor gain	6 (2.8)	6 (2.8)
HER2	Overall	26 (12.1)	26 (12.1)
	Receptor loss	21 (9.8)	21 (9.8)
	Receptor gain	5 (2.3)	5 (2.3)

Changes of ER, PR and HER2 receptor profiling revealed 6% (6.9%), 20% (26.5) and 12.1% discordance rates for ER, PR and ER, respectively. In total, 13 cases showed a receptor gain and 69 cases showed a receptor loss when comparing t-CNB and LNM (applying the ≥1% cut-off).

**Table 4 cancers-14-01863-t004:** Competitive analysis of receptor changes for ER, PR and HER2.

LNM/t-CNB	Estrogen (≥1%)			Progesterone (≥1%)			HER2		
Parameters	no Loss/Gain	Loss/Gain	*p*-Value	no Loss/Gain	Loss/Gain	*p*-Value	no Loss/Gain	Loss/Gain	*p*-Value
Median age [years] (IQR 1)	62 (50–73)	53 (50–59)	<0.01	61 (48–72)	63 (51–75)	0.14	61 (49–72)	63 (51–74)	0.65
Ki 67—LNM [%] (IQR 1)	10 (2–30)	40 (1–80)	0.11	10 (2–30)	5 (1–75)	0.26	10 (2–40)	7 (1–33)	0.75
Ki 67—t-CNB [%] (IQR 1)	15 (7–40)	40 (7–66)	0.18	15 (5–40)	28 (10–40)	0.91	15 (5–35)	3 (14–67)	0.01
Pathological tumor stage [*n* (%)]									
pTis	11 (5.6)	3 (23.1)		10 (6)	4 (11.6)		12 (6.7)	2 (7.7)	
pT 1	89 (44.9)	2 (15.4)		75 (44.6)	16 (37.2)		78 (43.6)	11 (42.3)	
pT 2	69 (34.8)	5 (38.5)	0.11	56 (33.3)	17 (39.5)	<0.01	62 (34.6)	8 (30.8)	0.68
pT 3	21 (10.6)	2 (15.4)		23 (13.7)	0		18 (10.1)	5 (19.2)	
pT 4	7 (3.5)	1 (7.7)		4 (2.4)	4 (9.3)		8 (4.5)	0	
Neoadjuvant therapy [*n* (%)]									
No therapy	137 (69.5)	8 (61.5)		118 (70.7)	27 (62.8)		125 (70.2)	15 (57.7)	
NACT 2	41 (20.8)	4 (30.8)	0.7	36 (21.6)	9 (20.9)	0.23	34 (19.1)	10 (38.5)	0.63
Endocrine therapy	19 (9.6)	1 (7.7)		13 (7.8)	7 (16.3)		19 (10.7)	1 (3.8)	

Competitive analysis of receptor changes showing that a change of ER occurred significantly more often in younger patients and was associated with an increased Ki67 index. HER2 change was significantly correlated with a decreased Ki67 index in t-CNB, indicating a possible switch towards luminal A tumor characteristics. Tumor stage was not significantly associated with ER and HER2 heterogeneity. PR changes appeared to be correlated with tumor stage. There were no significant associations between a receptor change (ER, PR, HER2) and therapy-related parameters. ^1^ IQR: interquartile range; ^2^ NACT: Neoadjuvant chemotherapy.

## Data Availability

The data presented in this study are available on request from the corresponding author. The data are not publicly available due to privacy and ethical reasons.

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
