# Peer review of "Heterogeneity between Core Needle Biopsy and Synchronous Axillary Lymph Node Metastases in Early Breast Cancer Patients—A Comparison of HER2, Estrogen and Progesterone Receptor Expression Profiles during Primary Treatment Regime"

_cancers, 2022, doi:10.3390/cancers14081863_

Round 1
Reviewer 1 Report
The manuscript by Weydandt et al. titled “ Heterogeneity between core needle biopsy and synchronous axillary lymph node metastases in early breast cancer patients - comparison of HER2, estrogen and progesterone receptor expression profiles during primary treatment regime” describes a robust retrospective cohort of breast cancer samples for discordance between tumour biopsies and lymph nodes. They show immunohistochemical staining discordances between primary tumour biopsy and the lymph nodes, especially for progesterone receptor, which could lead to potential differences in patient treatment strategies.
- While the paper is highly interesting it is difficult to judge how these findings fit with other studies? A review by Yao et al Medical Oncology 2014 has a section describing this topic. How does this study compare with some of these articles regarding marker discordance?
- Is there any relationship between the discordance observed and the number of lymph nodes? Did the authors stain all the lymph nodes collected and where there any differences between the nodes if more than 1 lymph node was positive?
- Section 3.3 – how did the authors arrive to their results? They only observed receptor gain in 13 cases, however on line 230 and in figure 3 they report an increase in the Luminal A subtype, and thus receptor gain, by 19 cases. This is a confusing section.
- The discussion is well written and not overinterpreted. A thoroughly enjoyable discussion to read.
- Regarding the sentence about de-differentiation of cancer cells (lines 313-314), how can the authors really make this claim? Their analysis cut-offs for marker expression are between 1-10%, indication that a proportion of tumours have between 90-99% of cancer cells negative for ER, PR and Her2 expression. Thus, it could be that maybe the triple negative cells are the more motile cancer cells and are the cells that metastasise to the lymph nodes. This sentence should be carefully re-worded. It would be nice if there was data from another epithelial marker that identifies these triple negative cells showing positivity in the primary tumour and the lymph nodes to indicate which cell types are metastasising.
Reviewer 2 Report
This manuscript is well written and tries to describe the heterogeneity of hormones receptor status and HER2 between core needle biopsy of primary breast cancer tissues and the paired axillary lymph node metastasis (LNM) at primary surgery. In particularly the fact that a clonal selection for more aggressive phenotype could be selected during metastatic event of the lymph nodes. Changes are reported not to correlate with more aggressive phenotype of the primary tumor, as consequence detection of receptors in the lymph nodes could help in better identifying patients at more risk at earliest stage.
Even though the authors define the LNM synchronic the primary surgery occurs with a delay of time; no indication about this time frame is reported in the manuscript. Moreover, 30% of all patients received neoadjuvant treatments: 46 chemotherapy (21.4%); 20 endocrine therapy (9.3%).
Only 149 (69.3%) received no treatment. An overview of the data (without performing statistic) gives the impression that no significant difference would be observed if only the untreated patients would be analysed at least for ER and HER2: ER loss/gain observed in 9 over 145 (5.5%) in non treated patients; PR loss/gain observed in 27 over 145 (18.6%) in non treated patients; HER2 loss/gain observed in 15 over 140 (10%) in non treated patients.
It is well known, that PR detections are much more sensitive to any kind of pre analytical effect e.g. time to fixation, temperature etc. No surprise that the higher values in loss/change are observed for PR.
The median age was 61 years (31-85y) letting assume that up to 40% of the patients could be premenopausal. It is well know that menstrual cycle may influence not only ER and PR status but also the HER2 status in premenopausal patients. e.g. Am J Pathol. 1999 Nov;155(5):1543-7. doi: 10.1016/S0002-9440(10)65470-3. Fluctuation of HER2 expression in breast carcinomas during the menstrual cycle. S Ménard et al. Observed Changes could in premenopausal patients could also be derived from the menstrual cycle effect.
All detections are performed by immunohistochemistry and no information regarding the inter-pathologist and intra-pathologists variations is reported. Significant results are obtained only after dicothomization with the threshold at 1 or 10%. It is known that inter and intra observe variations are mainly observed in this low range.
Even though the theory that most aggressive phenotype could be selected in infiltrating the lymph nodes, this manuscript is underpowered and does report any strong evidence of this effect.
Reviewer 3 Report
The purpose of this study was to present a comprehensive comparison of receptor expression profiles between pre-surgical t-CNB and synchronous axillary lymph node metastases at primary diagnosis, revealing therapeutically relevant aspects of tumor heterogeneity in breast cancer patients. This study demonstrates that the discordance rates between t-CNB and axillary LNM were 12% for HER2, 6% for ER, and 20% for PR. Receptor discordance appears to occur in the primary setting already. Hence, not only receptor profiles of the tumor tissue but also of the synchronous axillary LNM should be considered for treatment choice. These findings might add strength to the theory that therapy causes selective pressure, which could lead to a receptor change, in most cases a receptor loss. However, their data and explanation raise some questions and critics.
- The biggest drawback of this study is the small number of patients. In particular, in the case of TNBC, there are only 17 patients, so it seems unreasonable to see a statistical difference with such a small number of patients.
- In order to conclude that receptor profiles not only in t-CNB but also in axillary LNM are necessary, it is ideal to perform a biopsy of both primary tumor and LN at the time of initial diagnosis to confirm receptor profiles. The receptor profile through the axillary LNM at the time of surgery may have changed the luminal type due to neoadjuvant therapy. Therefore, it is necessary to mention whether there are any patients who have had axillary LN biopsy at the time of the first diagnosis and if there is a difference in receptor profiles from the primary tumor.
- To show that there was no change in receptor profiles due to neoadjuvant therapy, there is a method to check the receptor profile for primary tumor after surgery. Any data on this? If there is, it is recommended to add it to the manuscript.
- According to the analysis results, the author argued that it was not related to receptor change and neoadjuvant therapy, but according to table 3, in the case of HER2 or ER, the proportion of patients who received neoadjuvant CTx was 1.5 to 2 times higher in the group with receptor change. The statistically insignificant results are thought to be due to the small number of patients. It is recommended to group neoadjuvant CTx and endocrine therapy in one group and compare them with patients who do not receive neoadjuvant therapy.
- Even if there was no change in treatment because it was a retrospective study, it was recommended to retrospectively analyze the prognosis of patients who would have been good for hormone therapy due to positive findings among ER/PR/HER2 in axillary LN results although it was TNBC on t-CNB at the time of the first diagnosis.
Reviewer 4 Report
I read the manuscript with interest. The topic is of great interest, a retrospective study on a cohort of 215 breast cancer patients at different stages. The authors show their own experience of discordant immunohistochemical profiles of breast cancer. The role of axillary lymph node metastases has long been the subject of controversy in the literature, the work proposed by the authors does not appear to show an important novelty, confirming current knowledge. The heterogeneity of the tumor has been known for a long time. Most research reports on receptor mismatch mainly in the metastatic environment where patients have usually already received chemotherapy treatments.
General comment. In my opinion the article is too long, there are some parts with too many details (for example, introduction and materials, it could be abbreviated because I suppose this is according to well known guidelines, and also present in the tables, My recommendation is to reduce the article extension, improve the introduction paragraph.
However, we must consider the effort of the authors worthy of attention.
My main comments are as follows:
1. Expanding the introductory part, missing the discussion on axillary management, would make the research in its introductory part better, with which technique (Tc-99, indocyanin green, Superparamagnetic Iron Oxide?) Was the sentinel lymph node biopsied?
I would advise authors to revisit their literature search and at least add these works:
Cirocchi et al. New classifications of axillary lymph nodes and their anatomical-clinical correlations in breast surgery. World J Surg Oncol. 2021 Mar 29; 19 (1): 93. doi: 10.1186 / s12957-021-02209-2.
Mannell A et al. prospective study of receptor profiles in breast cancer and the ipsilateral axillary lymph node metastases measured simultaneously in treatment naïve cases. S Afr J Surg. 2020 Jun; 58 (2): 86-90. PMID: 32644312.
Georgescu et al Discordance Rate in Estrogen Receptor, Progesterone Receptor, HER2 Status, and Ki67 Index Between Primary Unifocal and Multiple Homogenous Breast Carcinomas and Synchronous Axillary Lymph Node Metastases Have an Impact on Therapeutic Decision
Appl Immunohistochem Mol Morphol. 2018 Sep; 26 (8): 533-538. doi: 10.1097 / PAI.0000000000000483. PMID: 28099174; PMCID: PMC6135467.
2. Both the objectives, methods and conclusions of this research are broad and radical.
3. The images are of good quality and standardized.
Round 2
Reviewer 2 Report
The authors have answered all the points raised in a satisfactory manner and have slightly modified the strenghthess of their results.
The authors should only add that all cases were evaluated by a single experience pathologist and if this expert revised all specimens / or the discrepant ones for this work.
Reviewer 3 Report
The manuscript is revised according to the reviewer’s comments.
Reviewer 4 Report
thanks to the authors for the responses to the comments I congratulate the authors on the improvements provided in the introduction and in the general writing of the paper